# Growth and Yield Models for Balsa Wood Plantations in the Coastal Lowlands of Ecuador

**Álvaro Cañadas-López [1,2], Diana Rade-Loor [3], Marianna Siegmund-Schultze [4], Geovanny Moreira-Muñoz [1], J. Jesús Vargas-Hernández [5] and Christian Wehenkel [6,*]**

[1] Carrera Ingeniería Agropecuaria, Facultad de Ingeniería Agropecuaria, Universidad Laica Eloy Alfaro de Manabí, C.P. 130301 Av. Eloy Alfaro, Chone, Ecuador

[2] Programa de Forestaría, Instituto Nacional de Investigaciones Agropecuarias (INIAP), Estación Experimental Tropical Pichilingue, C.P. 120501 Km 5 vía Quevedo—El Empalme, Pichilingue, Ecuador

[3] Centro de Investigación de las Carreras de la ESPAM-MFL (CICEM), Escuela Superior Politécnica de Manabí (ESPAM-MFL), Campus Politécnico Calceta, C.P. 130250 Sitio El Limón, Calceta, Ecuador

[4] Environmental Assessment and Planning Research Group, Technische Universität Berlin, Straße des 17. Juni 145, 10623 Berlin, Germany

[5] Posgrado en Ciencias Forestales, Colegio de Postgraduados, Montecillo, Texcoco C.P. 56230, México

[6] Instituto de Silvicultura e Industria de la Madera, Universidad Juárez del Estado de Durango, Boulevard Guadiana #501, Ciudad Universitaria, Torre de Investigación, Durango C.P. 34120, México

\* Correspondence: wehenkel@ujed.mx

**Abstract:** Balsa trees are native to neotropical forests and frequently grow on fallow, degraded land. Balsa can be used for economic and ecological rehabilitation of farmland with the aim of restoring native forest ecosystems. Although Ecuador is the world's largest producer of balsa, there is a lack of knowledge about production indicators for management of balsa stands in the country. The aim of this study was to develop growth and yield models (i.e., site index (SI) curves and stem volume models) for balsa plantations in the coastal lowlands of Ecuador. Balsa trees growing in 2161 plots in seven provinces were sampled. Here we present the first growth and yield models for the native, although underutilized, balsa tree. Three curve models were fitted to determine SI for balsa stands, differentiating five site quality classes. Eight volume models were compared to identify the best fit model for balsa stands. The mean annual increment was used to assess balsa production. The generalized algebraic difference approach (GADA) equation yielded one of the best results for the height–age and diameter–age models. The Newnham model was the best volume model for balsa in this comparative study. The maximum annual increment (i.e., for the best stand index) was reached in the second year of plantation. The fitted models can be used to support management decisions regarding balsa plantations. However, the models are preliminary and must be validated with independent samples. Nevertheless, the very fast development of the native balsa tree is particularly promising and should attract more attention from forest owners and politicians.

**Keywords:** mean annual increment; stand density index; site index; silvicultural models; volume; native tree; underutilized tree

---

## 1. Introduction

Many non-native tree species have been used around the world for productive plantations, ornamental purposes and for rehabilitation of degraded land, eventually contributing to the homogenization of landscapes, reducing native diversity, and with mixed effects on ecosystem services [1]. Rehabilitation of degraded land is a worldwide necessity and is increasingly gaining political attention. In the context of climate change and biodiversity loss, locally-adapted solutions



are needed. The balsa tree, *Ochroma pyramidale* (Cav. Ex Lam), is native to the neotropical forests and frequently grows on fallow and degraded land. It can be used for economic, ecological rehabilitation of farmland with the aim of restoring native forest ecosystems [2,3]. In 2008, Ecuador was the world's top producer of balsa wood, producing 89% of the balsa sold worldwide, followed by Papua New Guinea with 8%. The global trade in sawn kiln-dried balsa wood and semi-finished wood products (155,000 m$^3$) was worth an estimated US $71 million [4].

According to Cañadas [5], the economic value of potential forestlands could be increased by using light-demanding trees such as balsa to establish plantations and agroforestry systems. Balsa has considerable potential in secondary forests and on abandoned agricultural land: It grows rapidly, the wood quality meets the requirements of the industry, and the species appears suitable for plantations and agroforestry systems. It could meet increased future requirements of the industry, as production costs are low [6,7]. Balsa is known as a "frame species" because of the following properties: (1) Good survival and high growth rates in degraded conditions; (2) rapid canopy development, which suppresses highly competitive heliophilous weeds; and (3) it provides food and perches that attract seed dispersal fauna [8]. These characteristics are associated with improved soil fertility [9], rehabilitation of degraded areas, and forest restoration [2]. Balsa plantations are deemed an economic alternative for the Ecuadorian coastal region because of the very light wood and versatility for adaptation to the needs of local small producers and timber exportation [10]. These characteristics contribute to the diversification of forestry production in smallholder farming systems [11].

The main components that influence the quality of plantation sites are climate and soil. In forest studies, site quality has been evaluated with the help of different indices. In even-aged stands, site index (SI) is the most popular means of assessing site quality and is also used to evaluate tree growth and yield potential. SI relates tree height or diameter of a species to tree age. However, SI and height growth are sensitive to incidents in the stand history [12,13]. Finally, measuring volume production of trees is a difficult task, and easier methods are desirable [14–17].

In 2016, the Ecuadorian Forest Incentives Program (PROFORESTAL), implemented by the Ministry of Agriculture, Livestock, and Aquaculture (MAGAP), recorded a total of 48,533 ha of forest plantations with the following species: teak (*Tectona grandis* L.), 19,602 ha; melina (*Gmelina arborea* Roxb.), 10,467 ha; balsa, 8518 ha; pine, 4733 ha; and other species, in the remaining 5213 ha [18]. Growth models predicting balsa stand development would be useful to estimate the stand dynamics and thus sustain decision making in stand management. Nevertheless, management of balsa stands suffers from a lack of reliable quantitative tools for simulating stand dynamics [19]. In addition, little research has addressed production, silvicultural, and management aspects of balsa, such as SI, stand density, growth rate, and soil fertility [19], except for a previous study in which we modelled the SI of Ecuadorian balsa for both height and diameter on the basis of the Chapman-Richards model [7]. However, there are wide array of growth and yield models, each with advantages and disadvantages. These models differ in the type of data used and the method of construction [20].

The aim of this study was to develop the first growth and yield models, including SI curves and volume models, for balsa plantations in the coastal region of Ecuador, thus contributing to the development of further silvicultural and economic studies.

## 2. Materials and Methods

### 2.1. Study Area

Balsa plantations were evaluated in seven provinces in the coastal region of Ecuador (Figure 1).

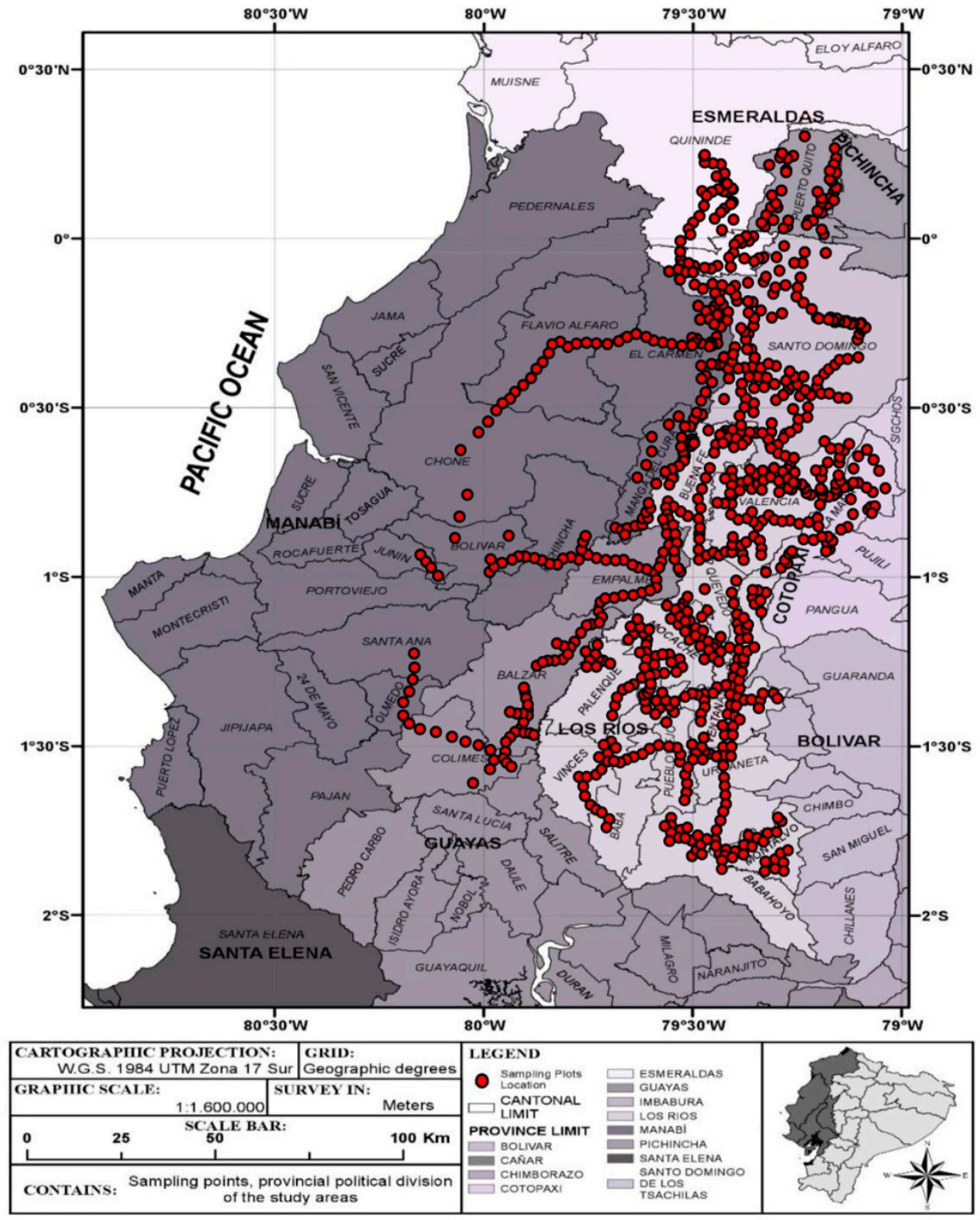

**Figure 1.** Spatial distribution of the balsa sampling plots in the coastal lowland provinces in Ecuador.

The climate data for the period 2009–2013 was taken from the Instituto Nacional de Meteorología e Hidrología (INAMHI) [21] meteorological yearbook. The number of plots sampled in each site was proportional to the surface area of each Balsa plantation. Table 1 shows the basic stand attributes of balsa plantations in the research region. Table 2 summarizes the annual precipitation, mean annual temperature, elevation, and the number of plots sampled by province and life zone, defined as evapotranspiration potential.

**Table 1.** Mean stand characteristics of the Balsa plantations by province in the coastal region of Ecuador.

| Province | | Density of Plantation (Trees ha$^{-1}$) | Age (Year) | Diameter at Breast Height (cm) | Total Height (m) | Basal Area (m$^2$ ha$^{-1}$) | Mean Annual Increment * (m$^3$ ha$^{-1}$) |
|---|---|---|---|---|---|---|---|
| Los Ríos | Average | 334.4 | 4.4 | 22.4 | 22.4 | 13.0 | 29.2 |
| | Maximum | 1506.0 | 10.9 | 44.1 | 40.6 | 26.0 | 228.1 |
| | Minimum | 90.0 | 1.1 | 2.9 | 2.5 | 0.3 | 2.2 |
| | Stand Err | ±712.5 | ±0.1 | ±1.1 | ±1.3 | ±0.6 | ±5.1 |
| Santo Domingo de los Tsáchilas | Average | 343.2 | 4.1 | 22.5 | 22.7 | 14.1 | 33.0 |
| | Maximum | 856.0 | 9.4 | 44.6 | 35.0 | 26.9 | 198.4 |
| | Minimum | 100.0 | 1.3 | 3.8 | 7.4 | 2.0 | 2.3 |
| | Stand Err | ±777.4 | ±0.1 | ±1.4 | ±1.6 | ±0.8 | ±9.9 |
| Cotopaxi | Average | 390.5 | 3.9 | 23.3 | 23.9 | 15.8 | 39.2 |
| | Maximum | 760.0 | 7.2 | 37.5 | 34.6 | 28.1 | 99.5 |
| | Minimum | 190.0 | 1.5 | 8.8 | 10.7 | 4.8 | 10.4 |
| | Stand Err | ±892.2 | ±0.2 | ±2.8 | ±2.9 | ±1.8 | ±15.3 |
| Manabí | Average | 280.7 | 5.1 | 21.9 | 23.1 | 11.6 | 22.9 |
| | Maximum | 800.0 | 10.8 | 35.2 | 34.3 | 22.5 | 43.9 |
| | Minimum | 90.0 | 1.4 | 6.8 | 7.4 | 2.2 | 3.0 |
| | Stand Err | ±280.7 | ±0.6 | ±3.8 | ±4.9 | ±1.7 | ±6.4 |
| Guayas | Average | 408.5 | 3.6 | 19.2 | 20.6 | 11.9 | 29.4 |
| | Maximum | 599.0 | 6.8 | 33.6 | 28.6 | 22.0 | 52.1 |
| | Minimum | 194.0 | 1.8 | 10.9 | 10.8 | 4.6 | 7.1 |
| | Stand Err | ±881.9 | ±0.2 | ±4.2 | ±3.4 | ±2.3 | ±13.4 |
| Esmeraldas | Average | 361.4 | 3.8 | 21.8 | 20.5 | 13.4 | 16.3 |
| | Maximum | 569.0 | 6.5 | 32.5 | 27.8 | 30.0 | 38.9 |
| | Minimum | 155.0 | 1.5 | 13.7 | 11.4 | 6.0 | 18.3 |
| | Stand Err | ±2842.5 | ±0.3 | ±4.2 | ±4.2 | ±4.9 | ±28.4 |
| Pichincha | Average | 337.0 | 3.9 | 24.6 | 20.2 | 11.6 | 23.1 |
| | Maximum | 493.0 | 6.8 | 32.6 | 27.6 | 19.2 | 37.4 |
| | Minimum | 210.0 | 1.8 | 15.3 | 9.9 | 4.0 | 7.4 |
| | Stand Err | ±1160.5 | ±0.6 | ±5.9 | ±5.7 | ±3.3 | ±9.8 |

* Note: Mean annual increment is the total growth of a stand up to a given age divided by the total age.

**Table 2.** Province, general conditions, and number of plots in the balsa plantations under study in the coastal region of Ecuador.

| Province | Total Annual Precipitation (mm) | Mean Annual Temperature (°C) | Elevation (Meters above Sea Level) | Number of Plots | Life Zone * |
|---|---|---|---|---|---|
| Los Ríos | 2000 | 25.2 | 30–300 | 1300 | TwF-TrF |
| Santo Domingo de los Tsáchilas | 2280 | 23.5 | 200–700 | 529 | Twf-TrF |
| Cotopaxi | 3019 | 24.0 | 300–600 | 117 | TmF-TrF |
| Manabí | 1000 | 24.6 | 20–300 | 100 | TdF-TrF |
| Guayas | 1189 | 26.7 | 30–100 | 56 | TdF-TrF |
| Esmeraldas | 2646 | 25.6 | 100–300 | 32 | TwF-TrF |
| Pichincha | 1998 | 25.0 | 300–700 | 27 | TwF-TrF |

* According to Cañadas [5]: Tropical dry forest, evapotranspiration potential 1–2 (TdF); tropical moist forest, evapotranspiration potential 0.5–1 (TmF); tropical wet forest, evapotranspiration potential 0.2–0.5 (TwF); tropical rain forest, evapotranspiration potential 0.1–2 (TrF).

## 2.2. Data Collection

The data was collected from a total of 2161 plots. Due to the planting density (Table 1), the sampled plot sizes varied from 500 to 5000 m$^2$ to yield a minimum of 40 balsa trees per plot. Each rectangular plot was georeferenced and coded, and the plantation age was recorded after consulting the plantation owner. Diameter at breast height (*DBH*) was measured with a caliper. The height (*H*) was estimated with a Haga hypsometer. Two consecutive measurements (2015–2016) were made.

### 2.3. Site Index Assessment

Mean dominant height ($H_d$) and diameter at breast height ($DBH_d$) were estimated as mean values for each sample plot of the 30 tallest balsa trees. In this study, different asymptotic models were applied to the height–age and diameter–age data to determine the best fitting model. Three different models were used to assess the SI: The model equations developed by Chapman-Richards [22] (Equation (1)), Hossfeld II cited in Peschel [23] (Equation (2)), and the generalized algebraic difference approach (GADA) (Equations (3) and (4)) [24]. All were computed in R software (version 3.3.3) (R foundation for Statistical Computing, Vienna, Austria).

$$H_d = a(1 - e^{(-bt)})^c \tag{1}$$

$$H_d = \frac{t(a_1 + a_2 t^2)}{a_3 + t^3} \tag{2}$$

where $H_d$ = dominant height (m); $t$ = age (year); $a_1$, $a_2$, $a_3$ = equation parameters. Polymorphic curves were developed from these curves, using the maximum height reached at 5 years of age as the reference [25] in this study. The guide curves obtained by Equations (1) and (2) were compared with the generalized algebraic difference approach (GADA) [26,27].

Owing to its flexibility, the three-parameter Chapman–Richards model is widely used in forest growth modelling and prediction, especially for developing site index models [28,29]. Parameter $a_1$ is the maximum value of this growth variable. $a_2$ is an empirical growth parameter scaling the absolute growth rate. The empirical parameter $a_3$ is related to catabolism rate [30]. The starting point of the curve is at the origin of axes, which is only useful if growth begins at zero [31]. The Hossfeld equation generates *S*-shaped curves giving a realistic growth of young trees and generates sensible curves when extrapolating outside the age and SI range of the data [32].

The advantage of GADA over guide curves and the algebraic difference approach (ADA) is that it produces dynamic site equations that are base-age-invariant (BAI) and have the path invariance property (PIP). GADA; thus, allows more than one parameter to be site specific. GADA site curves display "concurrent" polymorphism and variable asymptotes and use a variety of growth characteristics that operate across different sites [33].

For application of the GADA method, Equation (1) (Chapman-Richards) was selected to identify which parameters depended on site productivity. Thus, the selected parameters are expressed as site quality functions via the variable $X_0$. $X_0$ is an unobservable, independent variable that describes site productivity and reflects the management and ecological conditions and new parameters. The bi-dimensional equations ($H = f(t)$) were expanded to three-dimensional equations of the site index ($H_d = f(t, X)$), determining X from the initial site conditions ($t_0$ and $H_0$) [26]. The polymorphic curves with multiple asymptotes can be obtained by the GADA method with the following dynamic equation (Equation (3)).

$$H_1 = \left[ \frac{1 - e^{-b_1 t_1}}{1 - e^{-b_1 t_o}} \right]^{(b_2 + b_3/X_o)} \tag{3}$$

where $H_0$ is the dominant height at the initial age, $t_0$ and $H_1$ is the dominant height at age $t_1$. $H_0$ is derived for Equation (4).

$$X_0 = \frac{1}{2} \left\{ \ln H_0 - b_2 L_0 \pm \sqrt{[\ln H_0 - b_2]^2 - 4 b_3 L_0} \right\} \tag{4}$$

where $L_0 = ln[1 - exp^{-b_1 t_0}]$. Fitting this equation with the dominant height–age data enables estimation of the values of the global parameters $b_1$, $b_2$, $b_3$. The family of curves obtained by the GADA method are invariant relative to the reference age and invariant relative to the simulation path [27,34]. The mean structure (determined with the growth equation) and the mean error structure (determined by the auto regression model) were fitted simultaneously with the GADA package, in *R* (version 2.2.2) (*R* Foundation

for Statistical Computing, Vienna, Austria [35] (Development Core Team 2018). The procedure was also applied to $DBH_d$ in order to generate the three-mean dominant $DBH$–age models (Equations (1)–(4)).

Fitting the performance of the $H_d$ and $DBH_d$ models was based on comparison of curves. The bias and the root mean square errors (RMSEs) were calculated from the residuals obtained during the fitting stages. Graphical analysis was carried out by (1) superimposing the fitted curves, (2) plotting residuals against the values predicted by the model, and (3) analyzing changes in bias and RMSEs for the different age groups.

## 2.4. Tree Volume Assessment

A total of 532 dominant balsa trees (diameter average 23.1 ± 0.5 cm, height average 15.5 ± 0.6 m) were selected and destructively sampled in seven provinces with a total of 76 trees per province. The $DBH$ and $H$ were measured before and after felling the trees, respectively. A diameter tape was used to measure the diameter (di) over the bark on the ground and at different heights: 0.3 m, 2.3 m, and every 2.0 m along the stem to the top. The total tree volume of each tree ($V$, m$^3$) was calculated by measuring the diameter at each end of the section ($d_i$ and $d_{i+1}$) and the length of sections (l) of the felled specimens using the following formula [16]:

$$V = \frac{1}{3}\pi \left[ \left(\frac{d_i}{2}\right)^2 + \frac{(d_i d_{i+1})}{4} + \left(\frac{d_{i+1}}{2}\right)^2 \right] \tag{5}$$

Several commonly-used volume estimation models [36] were tested to determine the best regression model for $V$, $DBH$, and $H$ for the balsa stands (Table 3). The parameters to be determined are $a$, $b$, $c$, and $d$. The generalized method of moments (GMM), implemented in SAS/ETS® [37], was used to fit the models.

**Table 3.** Models tested for fitting volume equations for balsa trees by using diameter at breast height ($DBH$ in cm) and total tree height ($H$ in m) of dominant trees.

| Model | Expression | |
|---|---|---|
| Schumacher-Hall (allometric) [38] | $V = a \times DBH^b \times H^c$ | (6) |
| Spurr [39] | $V = a \times DBH^2 \times H$ | (7) |
| Spurr potential [40] | $V = a \times (DBH \times H)^b$ | (8) |
| Spurr with independent term [40] | $V = a + b \times DBH^2 \times H$ | (9) |
| Incomplete generalized combined variable [40] | $V = a + b \times H + c \times DBH^2 \times H$ | (10) |
| Australian formula [41] | $V = a + b \times DBH^2 + c \times H + d \times DBH^2 \times H$ | (11) |
| Honer [42] | $V = DBH^2 / (a + b/H)$ | (12) |
| Newnham [43] | $V = a + b \cdot DBH^c \times H^d)$ | (13) |

We estimated the goodness of fit of the volume models from the mean squared error (MSE), standard error (SE), and adjusted determination coefficient ($R^2_{Adj}$). Finally, the best model (of those tested) was selected by the Akaike information criterion (AIC), where lower AIC values indicate better performance [44]. Use of AIC or other criterions, such as the Bayesian information criterion, are a way of selecting the model that best balances (i) the true nature of the variability in the outcome variable and (ii) model generality [45].

## 2.5. Assessment of Balsa Production

The average density of the balsa stands in the study region was 300 trees ha$^{-1}$. The total volume was estimated by the modelled mean tree volume per age and SI multiplied by the indicated stand density. Mean annual increment (MAI) and periodic annual increment (PAI) were calculated. MAI is the total growth of a tree or stand up to a given age divided by the total age. PAI is equal to the tangent slope to the growth curve and describes the growth rate at age t. The PAI is cumulated at maximum tangent slope, where the tangent and the growth curve pass through the origin [46].

## 3. Results

### 3.1. Balsa Site Index

The estimated parameters for Equations (1)–(3), and the fitting statistics are shown in Table 4. All parameters were significant at the 1% level, including the site-dependent parameter for each tree. The lowest values of RMSE corresponded to the GADA method, for both $H_d$ and $DBH_d$.

**Table 4.** Estimated values of parameters, $p$ values, and goodness-of-fit statistics for the three-mean height–age and diameter–age models for balsa plantations of the coastal region of Ecuador.

| Parameter | Model | Parameter | Estimated Value | Standard Error | $p$ Value | RMSE |
|---|---|---|---|---|---|---|
| Height–Age | Equation (1) | $a_1$ | 31.844 | 0.6646 | <0.001 | 13.79 |
| | | $a_2$ | 0.2973 | 0.0247 | <0.001 | |
| | | $a_3$ | 1.0767 | 0.0662 | <0.001 | |
| | Equation (2) | $a_1$ | 0.8036 | 0.0065 | <0.001 | 6.43 |
| | | $a_2$ | −0.1382 | 0.0001 | <0.001 | |
| | | $a_3$ | 0.1331 | 0.001 | <0.001 | |
| | Equation (3) | $b_1$ | 0.4207 | 0.1483 | <0.001 | 4.27 |
| | | $b_2$ | −4.6136 | 0.2719 | <0.001 | |
| | | $b_3$ | 20.395 | 5.4538 | <0.001 | |
| Diameter–Age | Equation (1) | $a_1$ | 33.607 | 0.7364 | <0.001 | 23.06 |
| | | $a_2$ | 0.3108 | 0.0372 | <0.001 | |
| | | $a_3$ | 0.7876 | 0.0696 | <0.001 | |
| | Equation (2) | $a_1$ | 1.3090 | 0.00003 | <0.0000001 | 8.53 |
| | | $a_2$ | −0.1886 | 0.00001 | <0.001 | |
| | | $a_3$ | 0.1792 | 0.00004 | <0.0000001 | |
| | Equation (3) | $b_1$ | 0.7989 | 0.0791 | <0.0000001 | 5.61 |
| | | $b_2$ | −9.7819 | 2.3562 | <0.0000001 | |
| | | $b_3$ | 41.234 | 0.2583 | <0.0000001 | |

Figure 2 shows the results of analysis of the bias and RMSE evaluation for the age classes for both $H_d$ and $DBH_d$. Estimation with the GADA method showed a distribution of bias around zero for both parameters with no consistent trend. For RMSE, both Chapman-Richards and Hossfeld equations exhibited a similar trajectory for all age classes. The Chapman-Richards and GADA model showed a low RMSE, especially for the juvenile ages, for $H_d$. The largest bias and RMSE dispersions were particularly observed in the age class 9 to 10 years due to the lack of data for these classes. The GADA method; thus, yielded good fit models.

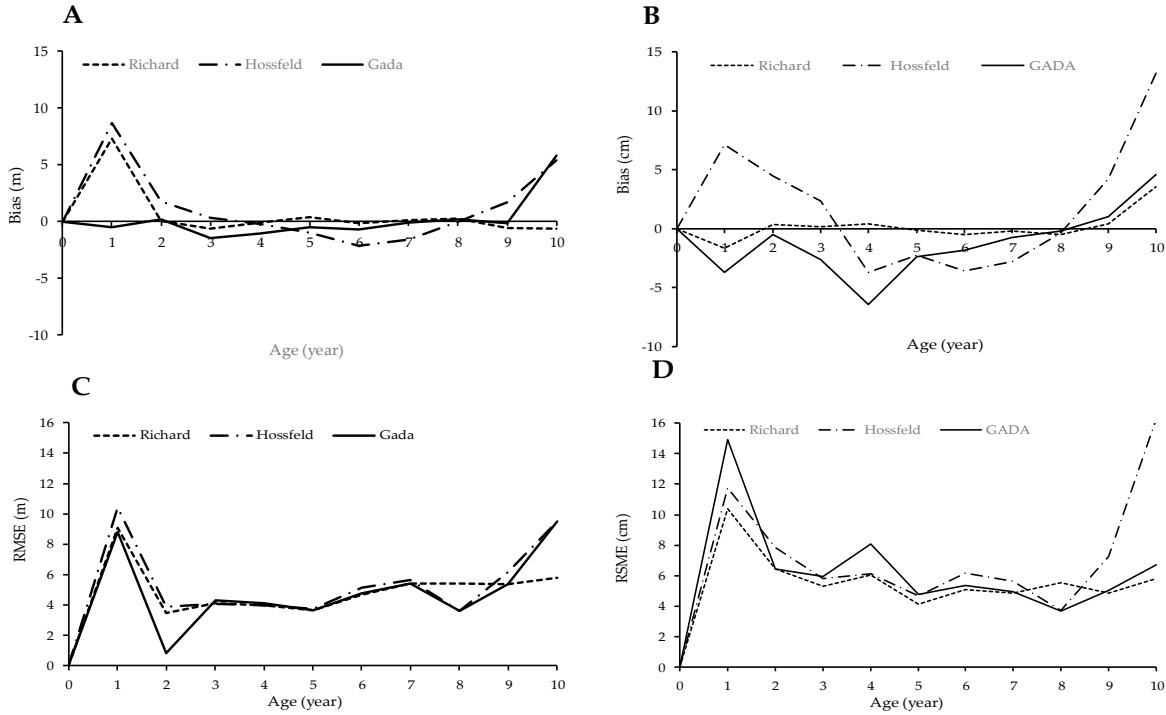

**Figure 2.** Bias (**A**,**B**) and root mean square error (RMSE; **C**,**D**) by age class for dominant height ($H_d$) (**A**,**C**) and dominant diameter at breast height ($DBH_d$) (**B**,**D**) estimated with the Equations (1)–(3) (Chapman-Richards, Hossfeld, GADA formulation, respectively).

With the GADA procedure based on the Chapman-Richards base model, both parameters $a_1$ and $a_3$ are dependent on-site quality and the error structure is included in the interaction procedure:

$$\text{Height}H_{d1} = H_{d0}\left[\frac{1-e^{-0.42017 \times t_1}}{1-e^{-0.42017 \times t_0}}\right]^{(-4.6136+20.3953/X_0)} \tag{14}$$

$$\text{Diameter}DBH_{d1} = DBH_{d0}\left[\frac{1-e^{-0.7989 \times t_1}}{1-e^{-0.7989 \times t_0}}\right]^{(-9.7919+41.2342/X_0)} \tag{15}$$

where $H_1$ is the predicted height (m) at age $t_1$ (years), and $H_0$ and $t_0$ represent the initial dominant height and age.

Height

$$X_0 = \tfrac{1}{2}\left\{\ln H_0 - 4.6136 \times L_0 \pm \sqrt{[\ln H_0 - 4.6136]^2 - 81.5812 \times L_0}\right\}$$
$$L_0 = ln\left[1-e^{(-4.6136 \times t_0)}\right] \tag{16}$$

Diameter

$$X_0 = \tfrac{1}{2}\left\{\ln H_0 - 9.7819 \times L_0 \pm \sqrt{[\ln H_0 - 9.7819]^2 - 164.9379 \times L_0}\right\}$$
$$L_0 = \ln\left[1-e^{(-9.7819 \times t_0)}\right] \tag{17}$$

Balsa site index was divided into five classes. Class one (I) represented the best site quality and class V the poorest quality both for *H* and *DBH*. The GADA model was used to fit SI curves for $H_d$ (from 34.6 to 15.3 m) and $DBH_d$ (from 40.3 to 15.6 cm) at a reference age of five years. The fitted curves follow the same trends. The curves did not perform better for any particular province (Figure 3).

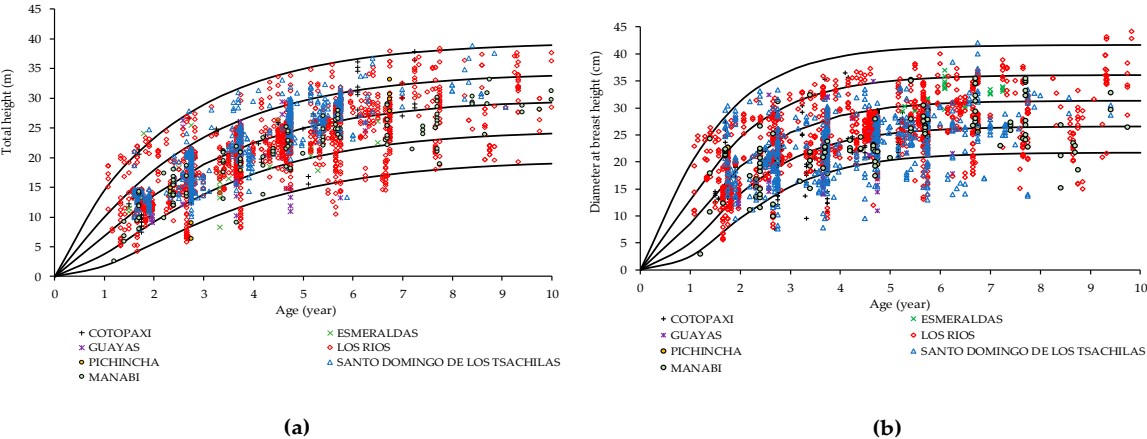

**Figure 3.** Site index curves for five site class for balsa in the coastal lowlands of Ecuador. (**a**) Total height (14, 19, 24, 29, and 34 m) and (**b**) diameter at breast height (20, 25, 30, 35, and 40 cm), using the GADA model (for a reference age of five years).

### 3.2. Volume Models

The outcome of the fitted regression of the eight-volume function estimation for balsa is given in Table 5.

**Table 5.** Goodness of fit statistics of the models predicting the volume (in m$^3$ for diameter at breast height over bark of 5 cm or more) of balsa plantations in the coastal lowlands of Ecuador.

| Model | MSE | R²$_{Adj}$ | Parameter | Estimator | SE | AIC |
|---|---|---|---|---|---|---|
| Newnham | 0.028 | 0.909 | a | 0.349727 | 0.0198 | −392 |
| | | | b | 0.000024 | 0.000005 | |
| | | | c | 2.343151 | 0.0629 | |
| | | | d | 0.760481 | 0.0518 | |
| Spurr with independent term | 0.028 | 0.907 | a | 0.322645 | 0.0124 | −380 |
| | | | b | 0.000042 | 0.0000003 | |
| Incomplete generalized combined variable | 0.028 | 0.907 | a | 0.279241 | 0.0528 | −379 |
| | | | b | 0.00375 | 0.00379 | |
| | | | c | 0.000041 | 0.0000008 | |
| Australiana formula | 0.028 | 0.907 | a | 0.282614 | 0.0572 | −378 |
| | | | b | −0.00006 | 0.000050 | |
| | | | c | 0.004089 | 0.00382 | |
| | | | d | 0.000043 | 0.000002 | |
| Schumacher-Hall (allometric) | 0.046 | 0.849 | a | 0.002402 | 0.000331 | −122 |
| | | | b | 1.100958 | 0.0465 | |
| | | | c | 0.819285 | 0.0738 | |
| Spurr potential | 0.046 | 0.848 | a | 0.002264 | 0.000281 | −121 |
| | | | b | 0.980555 | 0.0181 | |
| Honer | 0.058 | 0.809 | a | 973.3571 | 38.4457 | 1.56 |
| | | | b | −1995.99 | 822.1 | |
| Spurr | 0.081 | 0.733 | a | 0.000055 | 0.0000004 | 177 |

MSE = mean squared error; R²$_{Adj}$ = adjusted determination coefficient; SE = standard error; AIC = Akaike information criterion.

Half of the tested models provided good fits to the data with an adjusted $R^2$ > 0.9. For the Newnham, Spurr with independent term, incomplete generalized combined variable, and Australian formula models, the adjusted $R^2$ values were 0.9. According to the AIC criterion, the model that best

fits the characteristics of the balsa tree stands was the Newnham model (Equation (13)). However, this model overestimates the volume of trees with a *DBH* under 15 cm.

### 3.3. Mean Annual Increment

The optimal biological rotation was obtained for each SI at a reference age (Figure 4). The rotation lengths were three years for the best site, expanding to five years for the worst sites, when there were no other constraints. Considering a balsa tree density of 300 trees ha$^{-1}$, the MAI was 194.8 m$^3$ ha$^{-1}$ for the best SI, 86.6 m$^3$ ha$^{-1}$ for intermediate SI, and 27.7 m$^3$ ha$^{-1}$ for the lowest SI.

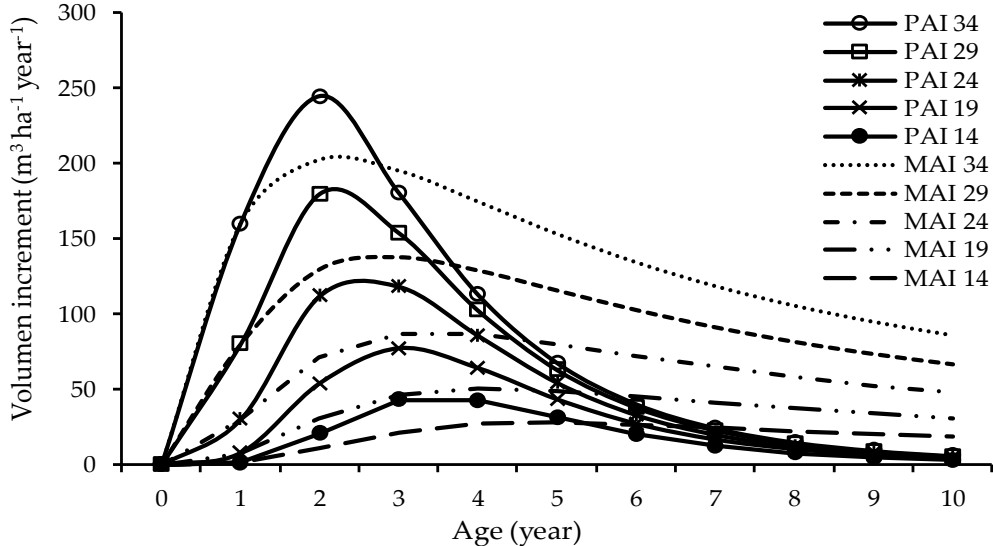

**Figure 4.** Relationship between mean annual increment (MAI) (dashed line) and periodic annual increment (PAI) curves (solid lines) for the volume increment at different ages and SI at five years. Coincidence of these two parameters can be considered to indicate the optimum biological rotation age with a density of 300 trees ha$^{-1}$ for the balsa plantations in the littoral region of Ecuador. The numbers associated with MAI and PAI correspond to the SI estimated by the height (14, 19, 24, 29, and 34 m) of mean dominant trees at five years. The volume was estimated with Equation (6) [38].

## 4. Discussion

### 4.1. Site Index Assessment of Height and Diameter

Comparison of the Chapman-Richards and Hossfeld models with the GADA model revealed that the latter provided better fits and site index curves for both $H_d$ and $DBH_d$ and generated polymorphic curves. These types of curves are comparable with those obtained by Cañadas et al. [36] for *Tectona*; Rodríguez-Carrillo et al. [47] for *Juniperus*; Díaz-Maroto et al. [48] for *Quercus* and Diéguez-Aranda et al. [49] for *Pinus*. The Chapman-Richards and Hossfeld models presented early asymptotes and greatly overestimated $H_d$ and $DBH_d$ at early ages. The Chapman-Richards models for balsa were used by Cañadas et al. [10] in Ecuador to determine the SI from both height and diameter. By contrast, the polymorphic curves obtained by the GADA method best described the growth of balsa, especially for plantations aged between one and three years old. Graphical analysis of bias and RMSE showed that the GADA model best described the biological parameters by passing the site index curves for the observed data of $H_d$ and $DBH_d$ age. These results are consistent with those obtained by Stankova and Diéguez-Aranda [50], who highlight the benefits of the GADA procedure, which provides the best model fits.

Gräfe [51] observed a surprisingly rapid growth of balsa at a young age within a secondary forest. This explains why balsa reaches its final height after only seven years. Under favorable environmental conditions this was about 20 m, and under unfavorable conditions, about 12 m. The final growth point

of *H* and *DBH* was not observed in our study areas, where balsa was cultivated in plantations of the same age.

According to Lamprecht [52], balsa (*Ochroma lagopus* (Cav. Ex Lam.) Urb) generally requires average annual precipitation of between 1500 to 3000 mm year$^{-1}$ and an average temperature of 22 to 27 °C. The edaphic demands for optimal growth of such forest plantations are exceptionally high. Optimal growth of balsa only occurs in deep soils of alluvial origin, with a sandy or slightly clayish texture, resulting from the weathering of rocks rich in bases. These edaphoclimatic conditions coincide with the best sites for growing balsa in the province of Los Ríos (Table 1), in the Ecuadorian coastal lowlands, as mentioned above. Our data differed from those obtained by Lamprecht [52], who recorded a height of 19 m and *DBH* of 33 cm at the age of five years in fertile alluvial soils of the Venezuelan western llanos.

In Panama, heights of 7.0 m were recorded for the Soberania site (precipitation of 2226 mm year$^{-1}$, 4.0 dry months), on the Los Santos site 5.8 m (1946 mm year$^{-1}$, 5.2 dry months) and 3.6 m in Río Hato (1107 mm year$^{-1}$, 6.7 dry months) for plantations aged two years [53]. These values are lower than those obtained in the present study.

Sites with rainfall equal or below 1000 mm year$^{-1}$ with seven dry months (here the province of Manabí in the tropical dry forest life zone, TdF-TrF with evapotranspiration ratio of 1–2) or precipitation of 3019 mm year$^{-1}$ and no dry months (province of Cotopaxi, Twf-TrF with a evapotranspiration ratio 0.2–0.5) were limiting conditions for balsa growth, indicated by $H_d$ = 15.3 m, $DBH_d$ = 15.6 cm to $H_d$ = 20.0 m, $DBH_d$ = 21.4 cm at five years. This is consistent with the observations made by Butterfield [54], Dalling et al. [55], Pearson et al. [56], Healy et al. [57], van Breugel et al. [58], and Azambuja et al. [59], who emphasized that (micro-) variations in water and soils locally affect the tree growth and survival of balsa.

Early successional species such as balsa generally show exceptional growth: Their crowns suppress weeds [54] and quickly improve the environmental conditions [60]. Balsa is characterized by high survival rates [61]. Nevertheless, de-Miguel et al. [19] observed high mortality in balsa, which affected by suboptimal environmental conditions in Bolivia and is considered a consequence of manual weed control. Such occurrences were not observed in the balsa plantations in our study region, as 61.5% of the weed control is carried out with chemicals in this part of Ecuador. Lamprecht [52] mentioned that no economic damage or diseases have been observed in plantations of *O. lagopus*, another species of balsa grown in tropical regions.

In this study, efforts have been made to develop god fitting balsa site index models for the coastal lowlands of Ecuador. However, these models can be improved in relation to dendro-ecological resolution, developing individual growth models used in other studies [13,62]. The inclusion of additional variables such as position, size of specific trees and spatial distribution could help to improve the understanding of the stand growth and production [13].

### 4.2. Tree Volume Assessment

Volume estimation plays a major role in productive forests. This is the first report of balsa volume models. In general, the Newnham model (Equation (13)) consistently provided adequate volume estimates, as demonstrated by the fitting statistics for balsa, indicated by the low MSE and AIC and high $R^2_{Adj}$ value. Newnham [43] proposed use of the Newnham model to estimate the volumes for *Pinus banksiana* Lamb., *Pinus contorta* Dougl., *Picea glauca* (Moench) and *Populus tremuloides* Michx in Alberta, Canada. In Ecuador, Cañadas et al. [36] used the Newnham model in a comparative study to describe the volume of *Tectona grandis* L. (teak) under agroforestry systems, and the model also provided the best fit to the data. This was also found by Hernández-Ramos [63] for *Eucalyptus urophylla* S.T.Blake. The balsa volume equation based on the Newnham formula (Equation (13)) overestimated smaller trees (0.34 m$^3$), because parameter *a* amounted to 0.34 (i.e., the smallest tree (*DBH* = 5 cm) would already have a volume of 0.34 m$^3$).

However, to date, the commercial volumes have only been determined by the simple, but less accurate, volume model, predicting tree volume as a function of *DBH*, height, and absolute form factor of 0.7 [64–67].

*4.3. Mean Annual Increment in Volume*

We chose to use the Schumacher-Hall (allometric) (Equation (6)) formula to calculate the volume and to estimate MAI and PAI, due to the better estimation of small tree dimensions. MAI and PAI coincided at three years of the balsa plantation growth as optimal biological rotation in the best site conditions. Therefore, the maximum rotation length increased rapidly with decreasing SI. These results are very different from those obtained by de-Miguel et al. [19], who proposed that the duration of the maximum rotation of balsa plantations in Bolivia would be one year if the size of trees and economic aspects are not considered. In the present study, the results indicated a three-year rotation for the best SI and five years for the SI. Harvesting at that age would take advantage of the low specific gravity of the young wood [68]. However, these data differ from those obtained by Longwood [69] who suggested rotation lengths of between four to six years for balsa plantations. Webb [70] suggested rotations of between seven to eight years. González-Osorio et al. [71] and Midgley et al. [4] determined optimal rotation lengths of between four to eight years. All the aforementioned researchers did not describe the method applied to determine the rotation age. However, de-Miguel et al. [19] assumed a rotation of five years in a study conducted in Bolivia using an individual-tree innovative approach.

Nevertheless, beyond technical forestry concerns, in the tropical lowland regions of Ecuador balsa is commercialized when a *DBH* of 18 cm or more is reached. This is due to the traders' prices for whole trees (10 to 15 US $ per tree). However, industry demands larger *DBH* (33–40 cm), doubling the tree value [71], and such diameters can be reached in the best SI at three years of age (Figure 2B).

The maximum variations in MAI of 202, 87, and 28 m$^3$ ha$^{-1}$ for good, medium, and poor sites, respectively, demonstrated that balsa behaves as a typical pioneer species and is highly sensitive to site conditions [4]. Wycherley and Mitchell [72] concluded that MAI followed a decreasing trend of this forest parameter at plantation establishment. By contrast, we have found that the maximum MAI was reached after two years for the best SI, four years for intermediate SI, and five years for the worst SI.

For the best SI, the MAI observed in the present study cannot be compared in quantity nor in rotation lengths with those obtained by Webb [70], who measured MAI in volume between 17 and 30 m$^3$ ha$^{-1}$ year$^{-1}$ in tropical regions at ages of seven and eight years old. However, plantation densities were not provided. Nevertheless, Howcroft [73] and Midgley et al. [4] mentioned that stands older than five to seven years were characterized by poor growth and wood quality. In Bolivia, de-Miguel et al. [19] predicted a mean annual commercial volume increment of 5.9, 10.4, and 12.9 m$^3$ ha$^{-1}$ year$^{-1}$ in five-year-old plantations, for poor, medium, and good sites, respectively; plantation densities were not presented because of the high mortality recorded.

## 5. Conclusions

In summary, forest models describe how trees grow and how forest structures are modified over time. The site index models for *DBH* and *H*, and volume and MAI, obtained in the present study must be validated with an independent sample for comparison of these forest parameters. Such validation could contribute to enable reasonable predictions to be made about balsa tree growth and stand development in the coastal lowlands of Ecuador. Moreover, balsa is an example of a native tree whose socio-economic and ecological potential is not yet fully known or exploited. Balsa is a pioneer species that thrives on abandoned land and is maintained with minimal input of labor, usually with a high survival rate and triggering forest succession. Despite the multiple attributes of this native species, it has unfortunately not yet been promoted by the Ecuadorian state reforestation program. However, the current environmental challenges associated with climate variation or change, deterioration of ecosystem services, and decline in biodiversity require several silviculture solutions.

These can be obtained through reforestation with fast-growing native species with high potential of biomass production, while at the same time recovering degraded areas.

**Author Contributions:** Á.C.-L., D.R.-L. and G.M.-M. conceived and designed the experiments, collected the data, and wrote the first draft; Á.C.-L., J.J.V.-H. and C.W. expanded the statistical analyses and M.S.-S. the argumentation lines; C.W. and M.S.-S. carried out a detailed revision of the paper.

**Funding:** This research received no external funding.

**Acknowledgments:** We thank the Director of the Portoviejo Experimental Research Station (EEP) of the National Institute of Agricultural Research (INIAP) and the INIAP General Director for providing the necessary logistics and resources in order to carry out this project.

**Conflicts of Interest:** The authors declare no conflict of interest.

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
