# Peer review of "Growth and Yield Models for Balsa Wood Plantations in the Coastal Lowlands of Ecuador"

_forests, doi:10.3390/f10090733_

Round 1
Reviewer 1 Report
The article describes and present growth and yield models for Balsa wood plantation in coastal lowlands Ecuador. The topic is interesting.
Minor comments
- The affiliations of authors refer 7 institutions but only 6 are listed. The institutions 4 and 5 listed are not affiliated at all.
- Row 34 the mean annual increment is reached in the third year … I guess you want to write maximum MAI is reached.. and also should be mentioned here this is for the best SI
- Figure 1 the state names are not readable
- Height and diameter are plotted against age class .. what is it?
- Mean annual increment is plotted against age in years .. why?
- Some of the values look really extreme (if the age class is years) some trees would have height 18 meters and diameter 20 cm after the one year of growth. Can you confirm these values?
Author Response
Please find the response on the attached file.

Reviewer 2 Report
In this paper, authors develop growth and yield models (site index curves) and stem volume models for Balsa plantations in Ecuador. Methods are appropriate, conclusions are developed based on achieved results and the goal was reached. However, there are many parts of the manuscript that will be understood only be a very limited group of experts in this area. In order to reach broader audience, I recommend as follows:
1) explain how growth and yield models can be developed, are there any other methods that can be used (this can be written in the Introduction);
2) explain shortly Chapman-Richards, Hossfeld and GADA method (in Methods); giving references does not help to understand differences between them;
3) why IAC criterion was used to evaluate volume models? Are there alternative methods?
4) it is not clear how mean annual increment was obtained (Table 2).
Moreover, I suggest the following major improvements:
5) Add information about the assessment of Balsa production in the Introduction and abstract; this issue is present in methods, results and discussion (the most developed part!)
6) Figure 1 – table below the figure is too small (unreadable)
7) Mark models I-V on Figure 5 (or add legend)
8) After table 2 all plots are analyzed together, and then in line 242 authors refer to regions again. However the statement is not justified by results (because results for other regions are missing). Table 1 and 2 show there are major differences between plantations in different provinces. More detailed analysis are welcome. In particular (but not comprehensive) please use colors in Figure 3 to mark plots from different provinces. Can you say, that some models (I-V) fit better to a particular province(s)? Is there a “correlation” between models and general province conditions (from Table 1)?
9) Explain the meaning of numbers (34, 29, 24, 19, 14) in Figure 4
Finally, a short list of minor corrections:
- Line 155 BHD ->DBH
- line 170 bthrough
- Table 2 Los Rios stand Err / Mean annual increment is missing +/-
Author Response

(The authors gave the same response as above.)

Round 2
Reviewer 2 Report
Authors have addressed all my comments and I am satisfied with the corrections they've made.